# Point Prevalence Survey of Antibiotic Use across 13 Hospitals in Uganda

**DOI:** 10.3390/antibiotics11020199

**Published:** 2022-02-04

**Authors:** Reuben Kiggundu, Rachel Wittenauer, JP Waswa, Hilma N. Nakambale, Freddy Eric Kitutu, Marion Murungi, Neville Okuna, Seru Morries, Lynn Lieberman Lawry, Mohan P. Joshi, Andy Stergachis, Niranjan Konduri

**Affiliations:** 1USAID Medicines, Technologies, and Pharmaceutical Services (MTaPS) Program, Management Sciences for Health (MSH), Kampala P.O. Box 920102, Uganda; jpwaswa@msh.org (J.W.); mmurungi@msh.org (M.M.); 2School of Pharmacy, University of Washington, Seattle, WA 98105, USA; rwitten1@uw.edu (R.W.); stergach@uw.edu (A.S.); 3Department of Global Health, University of Washington, Seattle, WA 98105, USA; hilman2@uw.edu; 4Sustainable Pharmaceutical Systems (SPS) Unit, Pharmacy Department, Makerere University School of Health Sciences, Kampala P.O. Box 10217, Uganda; kitutufred@gmail.com; 5Department of Pharmaceuticals and Natural Medicines, Ministry of Health, Kampala P.O. Box 7272, Uganda; nokuna6@gmail.com (N.O.); serumorries@gmail.com (S.M.); 6Overseas Strategic Consulting, Ltd., Philadelphia, PA 19102, USA; llawry@oscltd.com; 7USAID Medicines, Technologies, and Pharmaceutical Services (MTaPS) Program, Management Sciences for Health (MSH), Arlington, VA 22203, USA; mjoshi@msh.org (M.P.J.); nkonduri@msh.org (N.K.)

**Keywords:** point prevalence survey, antimicrobial stewardship, antibiotic use surveillance, antimicrobials, Uganda, hospital, private sector, global health security

## Abstract

Standardized monitoring of antibiotic use underpins the effective implementation of antimicrobial stewardship interventions in combatting antimicrobial resistance (AMR). To date, few studies have assessed antibiotic use in hospitals in Uganda to identify gaps that require intervention. This study applied the World Health Organization’s standardized point prevalence survey methodology to assess antibiotic use in 13 public and private not-for-profit hospitals across the country. Data for 1077 patients and 1387 prescriptions were collected between December 2020 and April 2021 and analyzed to understand the characteristics of antibiotic use and the prevalence of the types of antibiotics to assess compliance with Uganda Clinical Guidelines; and classify antibiotics according to the WHO Access, Watch, and Reserve classification. This study found that 74% of patients were on one or more antibiotics. Compliance with Uganda Clinical Guidelines was low (30%); Watch-classified antibiotics were used to a high degree (44% of prescriptions), mainly driven by the wide use of ceftriaxone, which was the most frequently used antibiotic (37% of prescriptions). The results of this study identify key areas for the improvement of antimicrobial stewardship in Uganda and are important benchmarks for future evaluations.

## 1. Introduction

Antimicrobial resistance (AMR) is a threat to global health and sustainable development, with adverse health and economic consequences, unless evidence-based efforts are implemented to control its emergence and spread [1,2]. The health and social consequences of AMR include increased morbidity and mortality, increased health care costs, and a projected negative impact on economic growth [3]. More than 700,000 people die annually from AMR, which is estimated to increase to 10 million annually by 2050 if decisive actions are not taken [4]. The potential exacerbating effects of the COVID-19 pandemic on the rise and spread of antimicrobial resistance have increased the urgency to address this problem [5,6,7,8]. AMR threatens the effective prevention and treatment of infections and undermines health gains globally as antimicrobials become less effective [9]. 

Numerous factors contribute to the emergence of AMR [10,11,12]. Among these factors is the irrational use of antibiotics in health care facilities. A 2015 situational analysis in Uganda showed a high prevalence of AMR to commonly used antibiotics [13]. Recent studies have also demonstrated the high prevalence of multidrug-resistant bacteria in Ugandan hospitals [14]. Further, pharmacokinetic and pharmacodynamic sex-based differences, in addition to gender roles, put females at higher risk of AMR [15]. As part of global efforts to contain AMR, the World Health Organization’s (WHO) Global Action Plan on AMR lists five strategic objectives for member countries to adopt and implement. A key aspect of the Global Action Plan is the surveillance of antibiotic use and consumption [16]. Recognizing the importance of antibiotic use surveillance, the Uganda National Action Plan on AMR (2018–2023) includes a strategic objective on surveillance of antibiotic use and consumption [17]. However, a key barrier to implementing this National Action Plan is the lack of current data and surveillance processes to monitor antibiotic use throughout the country, particularly within health facilities. To further strengthen antibiotic use surveillance at health facilities in resource-constrained countries, the WHO developed a standardized point prevalence survey (PPS) template and an associated package of tools which permit uniform collection and comparison of data within and among countries [18]. 

Recent efforts to measure antibiotic use in sub-Saharan African hospitals have been documented [19,20,21,22]. However, there are limited studies that utilize the standardized WHO PPS methodology for resource-limited settings, such as Ugandan hospitals [23,24,25]. This paper presents data from 13 hospitals in the context of a global health security agenda project for strengthening antimicrobial stewardship programs in low- and middle-income countries. This study was conducted as part of ongoing quality improvement approaches and efforts to build capacity for monitoring antibiotic use in health facilities, with a long-term goal of linkage to each hospital’s AMR containment program as well as national efforts to combat AMR [26,27]. These hospitals are depicted in Figure 1.

## 2. Results

Across the 13 included hospitals, de-identified data for 1077 patients was collected for analysis (Table 1). Of those patients, 609 (56.5%) were female, and the median age was 27 years old (IQR 10–38 years old). Patients were similarly distributed between maternal (28.8%), medical (22.2%), pediatric (22.5%), and surgical (26.3%) wards. Among all patients, at the time of data collection, 97.3% had a peripheral catheter present, 5.6% had a urinary catheter present, 0.5% were intubated, and 0.3% had a central catheter present. In terms of underlying health conditions of the included patients, at the time of data collection, 10.9% had malaria, 4.8% were malnourished, 4.3% were living with HIV, and 1.9% had tuberculosis. Approximately 66% of the included patients were in public hospitals and 34% were in private not-for-profit hospitals. Additional demographic and clinical characteristics of patients are summarized in Table 1 and the hospital characteristics are presented in Figure 1 and Appendix A.

### 2.1. Antibiotic Prevalence

Data were collected on 1387 antibiotics that were prescribed to patients in our study. Of these prescriptions, ceftriaxone was the most prescribed antibiotic (37%), followed by metronidazole (27%), gentamicin (7%), and ampicillin (6%) (Table 2).

#### 2.1.1. Prevalence by Indication

Among all the antibiotic prescriptions, the most common indication was for CAI (41.6%), followed by MP (29.1%), SP (23.0%), and HAI (6.3%) (Table 2). For all four indications, ceftriaxone and metronidazole were the two most prescribed antibiotics. For each antibiotic, the most common indication was CAI, except for metronidazole, which was most often prescribed for SP. 

#### 2.1.2. Prevalence by Diagnosis

Figure 2 summarizes the most prescribed antibiotic for each of the top five patient diagnoses within indications for CAI and HAI, which are the only two indication categories for which specific diagnosis data were collected. Among CAI indications, the most common diagnoses were clinical sepsis (20%), cellulitis, wound or deep soft tissue infection (19%), pneumonia (18%), gastrointestinal infections (13%), and symptomatic lower urinary tract infections (8%). Among HAI indications, the most common diagnoses were surgical site infections (41%), obstetric or gynecological infections (31%), cellulitis, wound or deep soft tissue infection (7%), intra-abdominal sepsis (7%), and pneumonia (5%). The three most frequently prescribed antibiotics for each of these five most common diagnoses in each indication are also summarized by antibiotic name and the WHO’s Access, Watch and Reserve (AWaRE) classification in Figure 2.

Indications for antibiotics were much more common for CAIs than HAIs. Clinical sepsis, cellulitis, pneumonia, and gastrointestinal infections were the most common CAI diagnoses for antibiotic use. Ceftriaxone and metronidazole are the most prescribed antibiotics at 37% and 27%, respectively. 

#### 2.1.3. Prevalence by Hospital

Prescribing patterns, in terms of antibiotic use prevalence, were similar across hospitals (Table 3). Among patients in 13 hospitals in Uganda, antibiotic use was common with 73.7% of patients receiving one or more antibiotics. Among hospitalized patients given an antibiotic, the mean number of antibiotics per patient ranged from 1.6 to 2.0 antibiotics per patient (Table 4). Public hospitals were significantly more likely to be associated with antibiotic use than private hospitals (OR 1.8, *p* < 0.01) (Table 5). Ceftriaxone and metronidazole were the two most prescribed antibiotics in all hospitals, except for Lacor hospital, where gentamicin and ampicillin were prescribed more frequently than ceftriaxone (Table 3). No patients were specifically treated based on antimicrobial susceptibility test laboratory results in any of the hospitals, as no hospitals in this sample regularly conducted sample collection or susceptibility testing as part of their surveillance activities.

### 2.2. Antibiotic Stewardship Indicators 

#### 2.2.1. Guideline Compliance

Among all antibiotics, only 30.1% (*n* = 423) were prescribed in compliance with the Uganda Clinical Guidelines 2016 [28]. Compliance with treatment guidelines was 30.9% of prescriptions in public hospitals and 29.7% in private not-for-profit hospitals (Table 6). In addition to Ruharo (which had 0% compliance but based on a small sample size of 10 total antibiotics), the lowest proportion of prescriptions in alignment with treatment guidelines was recorded at Kumi (15.6%, *n* = 12), and the highest proportion was at Kiwoko (54.5%, *n* = 24). Compliance with prescription guidelines for each antibiotic varied as well. Secnidazole, sulbactam, and tinizadole were always prescribed in accordance with guidelines, and metronidazole was prescribed with a relatively high rate of compliance (63%). All other antibiotics were not prescribed in accordance with guidelines more than half the time. Frequencies of compliance with treatment guidelines by hospital are summarized in Table 6 and compliance by antibiotic is summarized in Appendix A.

Overall, the indication for treatment was documented in patient record notes for 80.1% (*n* = 1373) of all prescriptions. St. Francis Naggalama recorded the highest proportion of prescriptions with the reason for prescribing antibiotics documented in the notes at 96.8% (*n* = 61), and Lacor hospital recorded the lowest at 64.3% (*n* = 153). Among all SP prescriptions, 1% (*n* = 3) of prescriptions were for 1 dose, 0.7% (*n* = 2) were multiple doses on day 1, and the remaining was for a longer duration, which was 301 (98.4%) multiple doses on more than 1 day. 

#### 2.2.2. WHO AWaRe Classification

Among all 1387 antibiotic prescriptions, 654 (47.2%) were from the Access group, 612 (44.1%) were in the Watch classification, and 9% were unclassified (Figure 3). The most prescribed Watch antibiotics were ceftriaxone (*n* = 519), ciprofloxacin (*n* = 45), azithromycin (*n* = 19), levofloxacin (*n* = 15), and erythromycin (*n* = 3). There were no antibiotic prescriptions in the Reserve group. The highest proportion of Access antibiotics was recorded at Lacor hospital (70.5%) and the lowest Access proportion was in Moroto (25.7%). Similarly, the highest proportion of Watch antibiotics was recorded at Moroto hospital (58.4%) and the lowest proportion at Lacor (20.5%). 

#### 2.2.3. Missed Doses

Among all antibiotics with more than 20 prescriptions, ampicillin-cloxacillin (30.4%, *n* = 24) had the highest proportion of courses that were not administered to patients, followed by metronidazole (15.1%, *n* = 57). By hospital, Lira hospital and Gulu hospital had the highest percentage of prescriptions that were not administered to patients (25.6%, *n* = 52; and 16%, *n* = 31, respectively), and five hospitals had prescriptions that were administered to all patients studied: Hoima, Kagando, Kiwoko, Kumi, and St. Francis Naggalama. The proportion of antibiotic courses that were administered by hospital are summarized in Appendix A.

#### 2.2.4. Route of Administration

Across all 1387 prescriptions, 11% (*n* = 157) were administered orally and 88% (*n* = 1230) were administered parenterally. A switch from parenteral to oral antibiotics was noted for 1.9% (*n* = 39) among all orally administered antibiotics. Among the antibiotics that were administered parenterally, 925 (75.3%) were administered intermittently, 302 (24.3%) were administered continuously, and 2 (<1%) were administered intramuscularly.

### 2.3. Antibiotics per Patient

Overall, 794 (73.7%) observed patients were on one or more antibiotics. By sex, 425 (69.8%) females and 369 (78.8%) males were on one or more antibiotics. Among patients on antibiotics, 302 (38%) were on one antibiotic, 440 (55%) were on two antibiotics, 44 (6%) were on three antibiotics, and two (<1%) were on four antibiotics. Among patients on any antibiotics, the mean number of antibiotics per patient was 1.55 (range 1–4). The mean number of antibiotics per patient ranged from 1.6 in St. Francis Naggalama Hospital to 2.0 in Ruharo Mission Hospital (*p* < 0.001). In publicly owned hospitals, the mean number of antibiotics per patient was 1.66, and in private not-for-profit hospitals, the mean number of antibiotics was 1.70 (*p* = 0.41) (Table 4).

Several characteristics were associated with significantly increased odds of being on antibiotics based on the univariate analyses (Table 5). These characteristics were being male, public hospital setting, specific hospital, patient HIV status, and patient malnutrition status (Table 5). Males had a 15% increase in the odds of antibiotic use, and public hospital settings had a nearly two-fold increase in antibiotic use than private, not-for-profit hospitals. Among the hospitals surveyed, there were up to nine-fold increases in antibiotic use. Lacor hospital had a 39% decrease in the odds of antibiotic use (*p* = 0.04). HIV and malnutrition patients had a nearly six-fold increase in antibiotic use (*p* < 0.001). Whether a patient was hospitalized in the previous 90 days did not increase the odds of antibiotic use. In the multivariate logistic regression, male sex (*p* = 0.003), HIV status (*p* = 0.003), and malnutrition status (*p* = 0.004) were significantly associated with odds of antibiotic use, while age and malaria status were not (Table 5).

#### Antibiotics by Sex

Among females, the characteristics associated with increased odds of antibiotic use were attendance at a public hospital (OR 1.60, *p* < 0.01), positive HIV status (OR 4.57, *p* = 0.04), malnourishment status (OR 5.21, *p* = 0.03), and attendance at the following hospitals: Hoima (OR 3.55, *p* < 0.01), Kumi (OR 5.49, *p* = 0.03), Masaka (OR 4.67, *p* < 0.01), and St. Francis Naggalama (OR 5.83, *p* = 0.02). Among males, the characteristics associated with increased odds of antibiotic use were attendance at a public hospital (OR 2.44, *p* < 0.01) and attendance at the following hospitals: Masaka (OR 6.51, *p* < 0.01), Moroto (OR 4.77, *p* = 0.01), and Soroti (OR 3.32, *p* = 0.03) (Appendix A).

## 3. Discussion

The observed prevalence of antibiotic use of 73.7% is somewhat similar to findings from certain PPS studies conducted in Kenya (67.7%) [29], Botswana (70.6%) [30], Ghana (60.5%) [31] and Jordan (75.6%) [32]. The prevalence of antibiotic use found in this study is high compared to what was found in high-income countries, but similar to findings in other low- and middle-income countries [33]. In contrast, a lower prevalence of antibiotic use was reported in other studies from Kenya (46.7%) [34]; Tanzania (44%) [35]; countries of Ghana, Uganda, Zambia, and Tanzania in the Global PPS (30–57%, with overall prevalence of 50%) [36]; Brazil (52.2%) [37]; Northern Ireland (46.2%) [38]; and Belgium (27.1%) [39]. Cross-national differences may be accounted for by factors such as varying disease burden, antibiotic use guidelines, and policies across countries, including ease of access to antibiotics, differences in patients’ characteristics, and types of hospitals. 

### 3.1. Antibiotic Use and Prevalence by Patient Characteristics

Both sex (identified by physical or physiological differences) and gender (defined by socially constructed roles) play important roles in AMR, based on sex differences in pharmacokinetics and pharmacodynamics and gender roles [15]. Body weight, blood volume, and fat distribution differences between the sexes have biological effects on how antibiotics are absorbed, distributed, metabolized, and eliminated. This is one of the reasons that females have a greater risk for AMR than males [40]. In this study, males had greater odds of being prescribed an antibiotic compared to females. In many societies where women are considered less valuable than men, gender determines the use of preventative measures and referral for more invasive therapeutic strategies [41]. To understand why, in the study, men were given more antibiotics would require a mixed-methods study to determine if the difference was a population-based difference in the proportion of males versus females, based on clinical indications or standards of care, lack of sex-specific antibiotic guidelines or due to other factors that suggest health inequity at the tertiary care level. Without better data, it is difficult to discern if there are sex differences in diagnosis indications at presentation, standards of care, or health care utilization (e.g., males may present later in the course of the disease) [42]. The WHO PPS does not allow for important data collection and analysis of sex and gender impacts on antibiotic use and AMR. Given WHO’s priority to equitably address AMR, the PPS should be updated to include these data in the PPS methodology [43].

### 3.2. Antibiotic Use Based on Uganda Clinical Guidelines and Hospital Setting

We found that most patients received multiple doses on more than one day, an average of 2.3 antibiotics administered per patient, contrary to WHO recommendations of single antibiotic prophylaxis [44]. In addition to the current practices of antibiotic use driving up the emergence of resistance, it also increases the cost of health care, with an increased risk of adverse drug reactions. There is a lack of guidance on the use of antibiotics in surgery in the Uganda Clinical Guidelines. The lack of guidelines on antibiotic use in surgery is a major concern for AMR emergence and needs to be addressed. Insufficient knowledge, prescriber attitudes, resistance to change, patient expectations exacerbated by scarce resources in countries such as Uganda and several other factors present additional challenges [45].

Use of parenteral antibiotics was very high (88%) compared to oral (12%) in the present study. These values are similar to the use of parenteral antibiotics in Indonesia (85.1%) and Pakistan (91.5%) [46,47]. The overuse of parenteral antibiotics (specifically, using parenteral antibiotics when not indicated or for longer than indicated) often increases costs of care, including costs associated with antibiotics, and nursing time and also increasing the duration of hospital stays; as such, overuse poses a challenge for infection prevention and control, especially in resource-constrained countries, such as Uganda. In this study, reasons for the indication for antibiotic prescription were written in the patient notes somewhat more frequently (80.1% of patients on antibiotics) compared to studies conducted in Indonesia (63.5%) and Pakistan (76.2%).

Our finding of poor adherence to Uganda’s Clinical Guidelines (30.1%), along with the high percentage of antibiotics used in the Watch category, further accentuates the need for effective antimicrobial stewardship programs in Uganda, in connection with progress toward universal health coverage [48]. Many factors could contribute to the low adherence to the Uganda Clinical Guidelines, including poor dissemination of the guidelines, lack of proper diagnostic stewardship in hospitals, such as in microbiology and radiology, and long turnaround times for laboratory results. Lastly, in many settings, prescriber preferences and behaviors are a major cause for non-adherence to the guidelines [49]. Similar factors have been described elsewhere as barriers to guideline adherence [50,51]. In contrast, in Tanzania, compliance with national guidelines was high at 84% and South Africa at 90.2% [52,53]. The overall low compliance with Uganda’s Clinical Guidelines could be a contributing factor to the observed high prevalence of Watch category antibiotics. In contrast, Lacor hospital, a private-not-for-profit, had a high adherence to the Uganda Clinical Guidelines, 39% less odds of antibiotic use, and hence higher use of the Access category antibiotics, partly attributed to the existence of its hospital antimicrobial stewardship program. The relatively high compliance to national guidelines in Tanzania and South Africa could partly be attributed to the implementation of antibiotic use policies and regulations, along with health care quality improvement initiatives [54,55]. A recent in-depth study of prescribers in Uganda found that stockouts of certain antibiotics, high patient load, prescriber’s years of experience, influence of pharmacies and pharmaceutical companies, patient demand, lack of ownership of the dangers of AMR and other factors contributed to poor prescribing practices [56]. Therefore, the Uganda Ministry of Health should strengthen and enforce policies to address the challenge of low compliance to guidelines in health facilities. 

Contrary to the general view that inappropriate antibiotic prescribing with respect to indication and quantity is higher in the private sector and based on a situational analysis [13], our findings showed no difference in prevalence of ceftriaxone prescribing, the percentage of guideline compliance, and mean number of antibiotics per patient, which was approximately the same when comparing the public and private-not-for-profit hospitals. Further, public hospitals were associated with 1.8 times higher odds of antibiotic use compared to private hospitals. This could be due to similar disease patterns and common prescriber behavior across both public and private practitioners. Health practitioners (including physicians, nurses, and pharmacists) commonly work in both private and public sectors. So, such working arrangements contribute to prescribing and dispensing behavior across both public and private sectors. 

### 3.3. Proportion of Prescribed Antibiotic Doses Not Administered to Patients

The finding of ampicillin-cloxacillin and metronidazole having a high proportion of doses not administered to patients (coded as “missed doses” in WHO PPS methodology) is not surprising since these drugs are administered at a higher frequency, i.e., every six hours and eight hours, respectively, compared to antibiotics like ceftriaxone, which is administered once a day. Public hospitals have a higher burden of patients and tend to use more higher-frequency dosing drugs. Other possible causes could be essential medicine stockouts, including certain Access group antibiotics, which means the antibiotics were not administered. The latter could also explain the high prevalence and, at times, lack of alignment with clinical guidelines for certain Watch group antibiotics, when the first-line Access group antibiotics ran out of stock [57,58]. These findings have implications for stewardship and may be associated with the emergence of resistance and have been associated with poor treatment outcomes [59,60]. Although causative factors for missed doses may vary between low- and middle-income countries and high-income countries, the use of continuous quality improvement plans has been found to be effective in addressing this challenge [61].

### 3.4. Route of Administration

The observed high proportion of parenteral antibiotics could point to overall stewardship challenges in health facilities. One possible explanation is the lack of adherence to the Uganda Clinical Guidelines. The observed low compliance with the guidelines (30%) could also explain the high proportion of parenteral antibiotics since most of the first-line drugs for treating common bacterial infections are oral medicines as per the guidelines. Additionally, implementing a hospital-parenteral-to-oral switch program requires regular patient review and availability of microbiology results to guide the switch to determine suitable oral medications. Human resource shortages could hinder regular patient review to switch antibiotics from parenteral to oral. Additionally, the lack of microbiology capacity in most hospitals may negatively affect the clinicians’ ability to empirically make decisions for the parenteral-to-oral switch. Lastly, the general belief that parenteral antibiotics work better than oral antibiotics could be a contributing factor [62]. Contributing factors and possible solutions to this challenge have been previously described [63].

### 3.5. WHO AWaRE Antibiotic Classification

Ampicillin-cloxacillin is listed as “not recommended” in the WHO AWaRe antibiotic classification database. It is considered as an inappropriate fixed-dose combination and is a problem not only in Uganda but in many low- and middle-income countries [64,65]. Ampicillin-cloxacillin was removed from the 2016 Uganda Clinical Guidelines. Reasons for its use could not be elucidated in our study, but it is concerning, given that it is among the top ten most consumed antibiotics nationwide in Uganda [66].

Ceftriaxone, a Watch antibiotic, was the most prescribed antibiotic for patients in these 13 regional hospitals and was used routinely for CAIs, MP, and SP. Ceftriaxone was used empirically as the first-line treatment for CAIs, contrary to Uganda’s Clinical Guidelines. Inappropriate use of ceftriaxone, which is a third-generation cephalosporin, can accelerate the emergence of AMR of multidrug-resistant organisms, increase treatment cost, and result in avertable adverse drug effects. In Uganda, one study reported 32% inappropriate use of ceftriaxone in nine health facilities while another study reported a high level of ceftriaxone use at a tertiary care, private not-for-profit hospital [67,68]. A possible explanation for the high prevalence of ceftriaxone could be the ease of use (single daily dose) coupled with its wide spectrum coverage, giving the prescriber a sense of broad-spectrum coverage for most infections encountered in clinical practice. 

Not using antimicrobial susceptibility testing when prescribing antibiotics in Uganda is also concerning. This situation is not unique to Uganda as only 2 out of 591 patients that received antibiotics in a PPS study conducted in Tanzania were specifically treated based on antimicrobial susceptibility testing results [52]. Among the challenges in antimicrobial susceptibility testing in sub-Saharan Africa are inadequate resources, weak supply chains for consumables for microbiological laboratory procedures, the timely turn-around of results for clinical decision-making (approximately 22 days in Uganda), and laboratory workforce limitations, such as staffing levels and training [69]. Underuse of culture sensitivity tests in hospitals is pervasive in resource-constrained countries. 

This study informs antimicrobial stewardship strategies for Ugandan hospitals by establishing measurable antimicrobial use targets, such as reducing the use of broad-spectrum antibiotics, complying with treatment guidelines and increasing the uptake of antimicrobial susceptibility testing. There is a need to develop and implement feasible strategies for complementing antimicrobial stewardship interventions such as in infection prevention and control and hand hygiene interventions for prevention of health care-associated infection that can reduce the need for antibiotic use. Additionally, the study can serve as a baseline to evaluate future antimicrobial stewardship strategies in Uganda’s hospitals. Areas of action could include establishing a funded national system for surveillance of antibiotic use in health facilities to inform antimicrobial stewardship interventions. There is also a need to strengthen implementation and enforcement of policies on the use of antibiotics. This could help reduce the inappropriate use of Watch category antibiotics and increase compliance with the Uganda Clinical Guidelines. For hospitals, survey findings can be used to develop specific continuous quality improvement plans to address the identified gaps. Areas of future research include understanding the potential enablers of implementing antimicrobial stewardship programs in Uganda; behavioral, sex, and gender impacts; and other factors that influence prescribing and consumer behaviors for antibiotics. Comprehensive solutions and multi-pronged approaches to tackle weak laboratory capacity in Uganda and sub-Saharan Africa in general have been extensively described elsewhere. Our study provides further evidence on the need for Uganda to secure appropriate investments to solve this vexing problem, which is an essential component of antimicrobial stewardship [70,71].

### 3.6. Limitations

Data collection took place over five months, rather than the WHO-recommended three-week window. A major cause of this limitation was the lack of capacity within health facilities to conduct antibiotic use surveys. The longer duration of our study may have resulted in some inconsistencies in antibiotic use patterns based on external factors, such as holidays and changes in disease transmission, including the COVID-19 burden. The PPS study design is restricted to assessing only inpatient antibiotic use. Consequently, antibiotics taken prior to hospitalization or those purchased externally and brought to the hospital are not recorded. The point-in-time nature of the PPS design further limits insight into seasonal patterns in antibiotic use. Even though the study included hospitals from different regions of the country as well as from government and private sectors, there are some limitations in the generalizability of our findings to other hospitals in Uganda and to hospitals in other countries with similar disease burdens and patient demographics. 

Our study has numerous strengths. Using the standard WHO PPS methodology allows for comparisons with future studies not only in Uganda but also in other East African settings and sub-Saharan African settings in general. Compared to other PPS studies performed in as few as 1–6 hospitals, the present study included 13 hospitals. Inclusion of public and private not-for-profit hospitals is another strength of the study along with the dispersed geographical coverage of the hospitals in Uganda. Our study built on the existing hospital medicines and therapeutics committee as the mechanism for stewardship interventions. Further, our findings go beyond measuring the use of antibiotics. Several study variables, such as the burden of surgical indications, reasons for antibiotic prescribing, and data on antibiotic use for specific sites, have the potential for additional utility to improve clinical decision making and ensure patient safety. Finally, we are the first to assess sex-disaggregated differences to understand inequities in antibiotic use among males and females by using the PPS methodology. 

## 4. Methods and Materials

All data were collected and analyzed based on the “WHO Methodology for Point Prevalence Survey on Antibiotic Use in Hospitals” version 1.1. The WHO Methodology for Point Prevalence Surveys, published in 2019, is intended to guide the collection of information on prescribing practices of antibiotics and other information relevant to treatment and management of infectious diseases in hospitalized patients [16]. Given challenges associated with data collection and high workload in resource-limited countries such as Uganda, the WHO PPS methodology has been developed with flexibility in mind. The data was collected to inform program activities and not submitted to WHO. Our study used the WHO PPS method over the Global-PPS method given our program’s consistent use of standardized WHO tools across countries including Uganda. The WHO-PPS methodology allows for paper-based data entry in hospitals with no access to computer devices.

### 4.1. Data Collection

#### 4.1.1. Setting

The study was conducted from December 2020 to April 2021 in 13 hospitals in Uganda. Six of the hospitals were public, government owned RRHs and seven were private not-for-profit hospitals. The public hospitals were chosen purposively based on ongoing involvement in related antimicrobial stewardship programs, and the private not-for-profit hospitals were chosen based on geographic location to cover four main regions of the country. Data were collected from all health facilities where our program is working to improve antibiotic use and no sampling of hospitals was done. Uganda has a decentralized health care system, with the lowest level being the Village Health Team and the highest being the National Referral Hospitals. General hospitals provide primary health care, covering the main disciplines of Medicine, Surgery, Pediatrics, and Obstetrics. The RRHs provide all the care provided by the general hospitals and provide specialized care in addition. 

#### 4.1.2. Data Collection

Data collection was conducted by staff from each hospital’s medicines and therapeutics committee. Training on using the WHO PPS standardized data collection methods, including the paper form, took place one day before the data collection began and included practice sessions. Data were included for all patients admitted to the ward before 8:00 am on the study day as well as all patients discharged on the study day. Data were excluded for patient types listed in the WHO PPS protocol, including all patients who were admitted after 8:00 am, patients in the palliative long-term care wards, patients currently undergoing radiological or surgical procedures, and patients who are still physically present in the ward but who have been officially discharged. Each ward was surveyed completely within one day to minimize impact of patients moving between wards. The variables collected were all core variables in the WHO PPS methodology for hospital, ward, patient, antibiotic, and indication categories of variables.

#### 4.1.3. Study Ethics and Approval

The Uganda Ministry of Health approved this study as part of ongoing technical assistance and implementation of the AMR national action plan [17].

### 4.2. Data Analysis

After data collection, de-identified data were entered into the WHO PPS Microsoft Excel-based tool by the study team for analysis. Data cleaning checks were performed for all 13 hospitals, and all data entry discrepancies were resolved using the original paper data collection forms. The data cleaning and entry was done as per the WHO methodology/protocol referenced above. 

Indication was grouped into four categories: community-acquired infection (CAI), hospital-associated infection (HAI), medical prophylaxis (MP), and surgical prophylaxis (SP). According to the WHO PPS methodology, infection is categorized as HAI if the date of onset is on: Day 3 onwards OR Day 1 or Day 2 AND patient transferred from another hospital OR Day 1 or Day 2 and patient discharged from a hospital (same hospital or another one) in preceding 48 h. Antibiotics were grouped according to 2019 WHO AWaRe classification of antibiotics for evaluation and monitoring of use [72]. Antibiotics in the Access category have a wide range of activity against common pathogens and show low resistance potential; Watch antibiotics have higher resistance potential and are intended to be key targets of stewardship programs and monitoring, and Reserve antibiotics are last-resort options when other alternatives have failed.

Descriptive statistics for the data are presented using mean ± standard deviation or median (IQR) for continuous variables. Results from categorical variables are expressed as proportions and the χ2 test or Fisher’s exact test was used, as appropriate, for comparisons. Univariate logistic regression to assess factors associated with increased odds of antibiotic use was conducted with the following prespecified factors: age (categorized as <2 years, 2–50 years, >50 years), sex, ward, hospital ownership, hospital facility, underlying condition status (HIV, TB, malaria, malnutrition, COPD), hospitalization within previous 90 days, indication category, and whether patient is a referral. This same univariate analysis was then performed disaggregated by sex as well. A multivariate logistic regression was also conducted which included pre-specified variables of interest: age, sex, HIV status, malaria status, and malnutrition status. Significance was evaluated at the alpha = 0.05 level. All analyses were performed using R Studio version 4.0.1 [73].

## 5. Conclusions

This is the first study in Uganda that was based on the WHO methodology for PPS on antibiotic use in hospitals. These data provide insights into the most common antibiotics used in 13 public and private not-for-profit hospitals throughout the country, including prevalence by patient characteristics, hospital, indication, and diagnosis. Notable findings were the high use of antibiotics among hospitalized patients, the high proportion of patients receiving parenteral antibiotics, the high prevalence of Watch antibiotics (particularly ceftriaxone), antibiotic use for SP nearly always spanning more than one day, low adherence to Uganda Clinical Guidelines, and sex differences in antibiotic use. These findings support the Uganda National Action Plan on AMR in identifying targeted strategies for improving antimicrobial stewardship. The estimates provided herein may also serve as a helpful baseline for the national antimicrobial stewardship technical working committee to promote and facilitate locally appropriate stewardship strategies in hospital settings and catalyze experience sharing and cross-fertilization. Additional research is needed to identify the drivers of inappropriate use in relation to disease burden and operational enablers of effective antimicrobial stewardship programs.

## Figures and Tables

**Figure 1 antibiotics-11-00199-f001:**
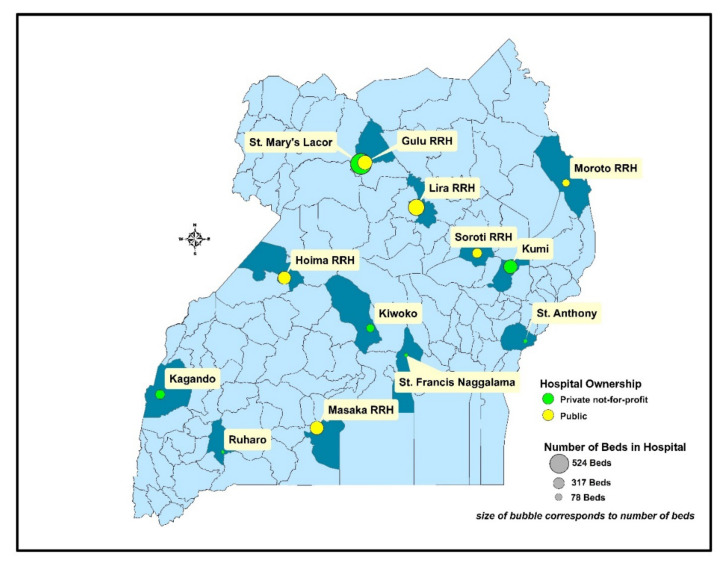
Geographic location of study sites and number of hospital beds.

**Figure 2 antibiotics-11-00199-f002:**
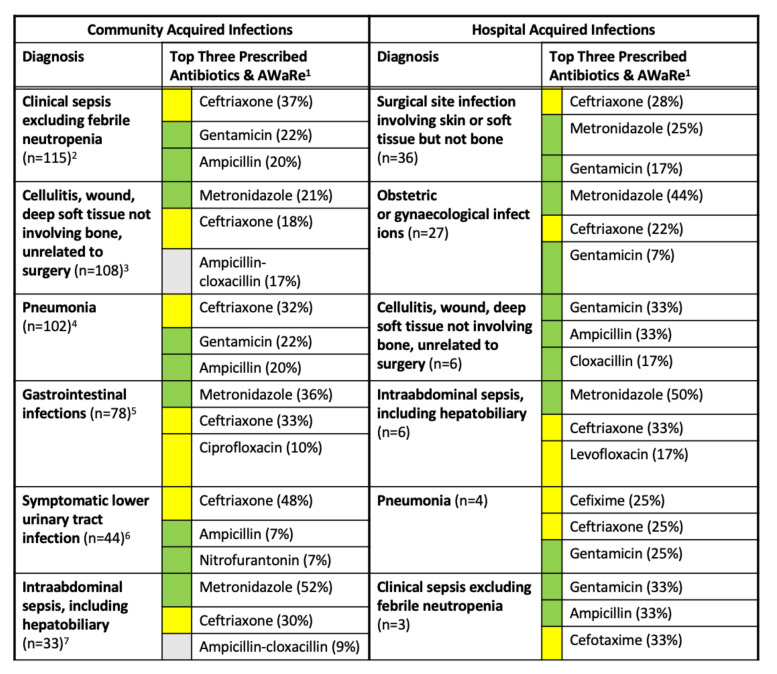
Top three antibiotics prescribed for various diagnoses and their AWaRe classification. Notes: ^1^ WHO AWaRe classification for antibiotics: ■ Access, ■ Watch, ■ Reserve, ■ not classified. ^2^ First-line treatment recommendation from the Uganda Clinical Guidelines 2016: ampicillin and gentamicin. ^3^ First-line treatment recommendation: cloxacillin. ^4^ First-line treatment recommendation: ampicillin and gentamicin (for children < 5yrs) or benzylpenicillin (for older children and adults). ^5^ First-line treatment recommendation: ceftriaxone and metronidazole, gentamycin (optional). ^6^ First-line treatment recommendation: nitrofurantoin or ciprofloxacin. ^7^ First-line treatment recommendation: ampicillin, gentamicin, and metronidazole.

**Figure 3 antibiotics-11-00199-f003:**
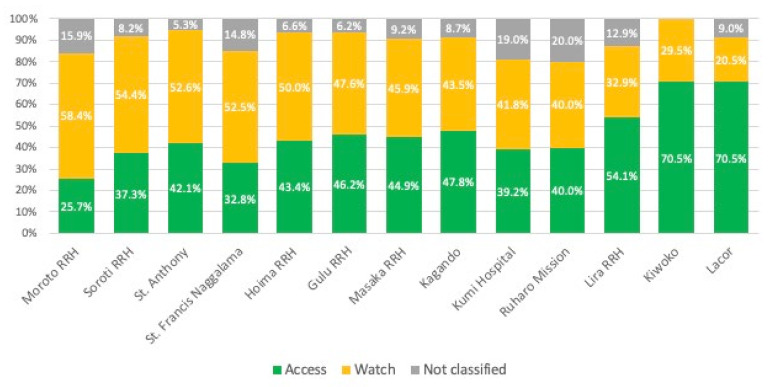
Prescriptions by AWaRe classification per hospital.

**Table 1 antibiotics-11-00199-t001:** Demographic and clinical characteristics of enrolled patients (*n* = 1077).

Variable	Number (Proportion) (*n* = 1077)
**Demographics**	
Female	609 (56.5%)
Male	468 (43.5%)
Age ^a^	27 (10–38)
**Hospital ownership**	
Public	706 (65.5%)
Private not-for-profit	371 (34.4%)
**Hospital**	
Gulu RRH	133 (12.3%)
Hoima RRH	103 (9.6%)
Kagando	61 (5.7%)
Kiwoko	43 (4%)
Kumi	47 (4.4%)
Lacor	168 (15.6%)
Lira RRH	119 (11%)
Masaka RRH	127 (11.8%)
Moroto RRH	99 (9.2%)
Ruharo Mission	6 (0.6%)
Soroti RRH	125 (11.6%)
St. Anthony	12 (1.1%)
St. Francis Naggalama	34 (3.2%)
**Ward**	
Maternal	311 (28.8%)
Medical	239 (22.2%)
Pediatric	243 (22.5%)
Surgical	284 (26.3%)
**Underlying patient condition**	
Central catheter	3 (0.3%)
Peripheral catheter	1049 (97.3%)
Urinary catheter	60 (5.6%)
Intubation	5 (0.5%)
Malaria	118 (10.9%)
Tuberculosis	20 (1.9%)
HIV	46 (4.3%)
COPD	12 (1.1%)
Malnutrition	52 (4.8%)

^a^ Age in years expressed as median (interquartile range). Abbreviations: COPD = chronic obstructive pulmonary disease; HIV = human immunodeficiency virus; RRH = regional referral hospital.

**Table 2 antibiotics-11-00199-t002:** Prevalence of antibiotic use by indication.

Antibiotic	All Prescriptions (*n* = 1387)	Community Acquired Infection (*n* = 577)	Hospital Associated Infection (*n* = 87)	Medical Prophylaxis (*n* = 404)	Surgical Prophylaxis (*n* = 319)
Ceftriaxone	513	183 (35.7%)	21 (4.1%)	177 (34.5%)	132 (25.7%)
Metronidazole	380	121 (31.8%)	26 (6.8%)	98 (25.8%)	135 (35.5%)
Gentamicin	119	70 (58.8%)	12 (10.1%)	22 (18.5%)	15 (12.6%)
Ampicillin	89	55 (61.8%)	5 (5.6%)	27 (30.3%)	2 (2.2%)
Ampicillin-cloxacillin	79	31 (39.2%)	4 (5.1%)	31 (39.2%)	13 (16.5%)
Ciprofloxacin	45	25 (55.6%)	2 (4.4%)	15 (33.3%)	3 (6.7%)
Cloxacillin	27	17 (63%)	1 (3.7%)	5 (18.5%)	4 (14.8%)
Amoxicillin	26	12 (46.2%)	0 (0%)	10 (38.5%)	4 (15.4%)
Azithromycin	19	15 (78.9%)	0 (0%)	3 (15.8%)	1 (5.3%)
Penicillin	16	10 (62.5%)	T0 (0%)	5 (31.3%)	1 (6.3%)
Levofloxacin	15	10 (66.7%)	4 (26.7%)	1 (6.7%)	0 (0%)
Other ^a^	59	28 (47.5%)	12 (20.3%)	10 (16.9%)	9 (15.3%)

^a^ Other category includes the following antibiotics: nitrofurantoin (*n* = 10), cefotaxime (*n* = 7), flucamox (*n* = 7), cef-sulbactam (*n* = 5), cefixime (*n* = 4), meropenem (*n* = 4), piperacillin-tazobactam (*n* = 4), sulbactam (*n* = 4), co-trimoxazole (*n* = 3), erythromycin (*n* = 3), ceftazidime (*n* = 2), amoxyclav (*n* = 1), doxycycline (*n* = 1), secnidazole (*n* = 1), tinidazole (*n* = 1), clindamycin (*n* = 1), cefazolin (*n* = 1).

**Table 3 antibiotics-11-00199-t003:** Prevalence of specific antibiotic use by hospital.

Antibiotic ^a^	Total (*n* = 1387)	Public Hospitals (All Regional Referral Hospitals)						Private Not-for-Profit Hospitals						
		Gulu (*n* = 144)	Hoima (*n* = 151)	Lira (*n* = 170)	Masaka (*n* = 203)	Moroto (*n* = 111)	Soroti (*n* = 157)	Kagando (*n* = 90)	Kiwoko (*n* = 44)	Kumi (*n* = 77)	Lacor (*n* = 150)	Ruharo Mission (*n* = 10)	St. Anthony (*n* = 19)	St. Francis Naggalama (*n* = 61)
**Ceftriaxone**	513	61	66	39	85	62	74	38	12	22	16	4	9	25
**Metronidazole**	380	43	53	52	62	14	47	28	10	16	33	3	6	13
**Gentamicin**	119	13	5	22	14	9	2	6	6	8	27	1	2	4
**Ampicillin**	89	5	2	11	14	6	5	5	5	6	28	0	0	2
**Ampicillin-cloxacillin**	79	8	7	19	9	11	8	6	0	3	1	0	1	6
**Ciprofloxacin**	45	6	5	8	4	3	4	1	0	2	11	0	0	1
**Cloxacillin**	27	0	2	0	0	0	0	0	4	0	20	0	0	1
**Amoxicillin**	26	6	4	4	1	0	0	4	6	0	1	0	0	0
**Azithromycin**	19	0	2	3	4	0	1	0	1	1	2	0	0	4
**Penicillin**	16	0	1	1	0	5	0	0	0	2	7	0	0	0
**Levofloxacin**	15	0	0	0	2	0	1	1	0	7	1	0	1	2
**Other ^b^**	59	2	4	11	8	1	14	1	0	10	3	2	0	3

^a^ Antibiotic AWaRe classification ■ Access, ■ Watch, ■ Reserve, ■ not classified. ^b^ Other category includes the following antibiotics: Nitrofurantoin (*n* = 10), cefotaxime (*n* = 7), flucamox (*n* = 7), cef-sulbactam (*n* = 5), cefixime (*n* = 4), meropenem (*n* = 4), piperacillin-tazobactam (*n* = 4), sulbactam (*n* = 4), co-trimoxazole (*n* = 3), erythromycin (*n* = 3), ceftazidime (*n* = 2), amoxicillin/clavulanic acid (*n* = 1), doxycycline (*n* = 1), secnidazole (*n* = 1), tinidazole (*n* = 1), clindamycin (*n* = 1), cefazolin (*n* = 1).

**Table 4 antibiotics-11-00199-t004:** Antibiotic use by hospital.

Care Setting	Mean Antibiotics per Patient (Range)
**Hospital ownership**	
Public	1.66 (1–4)
Private not-for-profit	1.70 (1–4)
**Hospital**	
Gulu RRH	1.631 (1–3)
Hoima RRH	1.727 (1–3)
Kagando	1.776 (1–3)
Kiwoko	1.792 (1–4)
Kumi	1.658 (1–3)
Lacor	1.667 (1–3)
Lira RRH	1.953 (1–3)
Masaka RRH	1.685 (1–3)
Moroto RRH	1.325 (1–4)
Ruharo Mission	2.000 (1–3)
Soroti RRH	1.646 (1–3)
St. Anthony	1.727 (1–3)
St. Francis Naggalama	1.625 (1–3)

**Table 5 antibiotics-11-00199-t005:** Associations of antibiotic use with characteristics of the study sample.

Variable	Antibiotic Use (*n* [%])	Univariate Model		Multivariate Model ^2^	
		Odds Ratio	*p*-Value ^1^	Odds Ratio	*p*-Value ^1^
**Age category**					
<2 years	102 (82.9%)	1 (reference)			
2–50 years	569 (71.7%)	0.65	0.01 *	0.63	0.08
>50 years	117 (75.8%)	0.52	0.16	0.70	0.28
**Sex**					
Female	425 (69.8%)	1 (reference)			
Male	369 (78.8%)	1.15	<0.001 *	1.57	0.003 *
**Hospital ownership**					
Private not-for-profit	245 (66.0%)	1 (reference)			
Public	549 (77.8%)	1.80	<0.001 *		
Hospital					
Gulu RRH	84 (63.2%)	1 (reference)			
Hoima RRH	88 (85.4%)	3.42	<0.001 *		
Kagando	49 (80.3%)	2.38	0.02 *		
Kiwoko	24 (55.8%)	0.74	0.39		
Kumi	38 (80.9%)	2.46	0.03 *		
Lacor	86 (51.2%)	0.61	0.04 *		
Lira RRH	86 (70.3%)	1.52	0.12		
Masaka RRH	114 (89.8%)	5.11	<0.001 *		
Moroto RRH	81 (81.8%)	2.62	0.002 *		
Ruharo Mission	5 (83.3%)	2.92	0.33		
Soroti RRH	96 (76.8%)	1.93	0.02 *		
St. Anthony	11 (91.7%)	6.41	0.08		
St. Francis Naggalama	32 (94.1%)	9.33	0.003 *		
Ward					
Maternal	219 (70.4%)	1 (reference)			
Medical	170 (71.1%)	1.04	0.86		
Pediatric	188 (77.4%)	1.44	0.06		
Surgical	217 (76.4%)	1.36	0.10		
**Underlying conditions**					
HIV (no)	689 (71.7%)	1 (reference)			
HIV (yes)	43 (93.5%)	5.65	0.004 *	5.90	0.003 *
TB (no)	705 (71.9%)	1 (reference)			
TB (yes)	18 (90%)	3.51	0.09		
Malaria (no)	667 (73.2%)	1 (reference)			
Malaria (yes)	85 (72%)	0.94	0.78	0.79	0.31
COPD (no)	752 (73.9%)	1 (reference)			
COPD (yes)	9 (75.0%)	1.06	0.93		
Malnutrition (no)	738 (72.9%)	1 (reference)			
Malnutrition (yes)	49 (94.2%)	6.06	0.002 *	5.78	0.004 *
Hosp in past 90 days (no)	696 (72.8%)	1 (reference)			
Hosp in past 90 days (yes)	65 (80.2%)	1.51	0.15		

^1^ Statistical significance is noted by an * for all relationships with *p* < 0.05. ^2^ Multivariate model includes: age, sex, HIV status, malaria status, and malnutrition status. Abbreviations: RRH = regional referral hospital; COPD = chronic obstructive pulmonary disease; HIV = human immunodeficiency virus; TB = tuberculosis.

**Table 6 antibiotics-11-00199-t006:** Antibiotics prescribed in compliance with Uganda Clinical Guidelines by hospital.

Setting	Guideline Compliance (*n*, %)
**Hospital ownership**	
Public	289 (30.9%)
Private not-for-profit	134 (29.7%)
**Hospital**	
Gulu RRH	41 (28.5%)
Hoima RRH	61 (40.4%)
Kagando	17 (18.9%)
Kiwoko	24 (54.5%)
Kumi	12 (15.6%)
Lacor	58 (38.7%)
Lira RRH	61 (35.9%)
Masaka RRH	67 (33%)
Moroto RRH	34 (30.6%)
Ruharo Mission	0 (0%)
Soroti RRH	25 (15.9%)
St. Anthnoy	5 (26.3%)
St. Francis Naggalama	18 (29.5%)

Abbreviations: RRH = regional referral hospital.

## Data Availability

Not applicable.

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
