# Peer review of "Point Prevalence Survey of Antibiotic Use across 13 Hospitals in Uganda"

_antibiotics, 2022, doi:10.3390/antibiotics11020199_

Round 1

Reviewer 1 Report

The authors did a point prevalence survey of antibiotics used across 13 hospitals in Uganda, and describe the amount of antibiotics used per patient at different hospitals in Uganda, antibiotics used by different underlying diseases, and antibiotics used by indication.  The data of this study may be valuable, as it provides basic data important for decreasing inappropriate and overuse of antibiotics. However, I have some major comments that should be considered: 

Major comments: 

  1. The authors include  data listing the number of antibiotics administered per patient, or the type of antibiotics used per underlying disease. However, in order to evaluate whether appropriate or misused antibiotics were administered, which is in fact more important than just giving data on the total amount of type of antibiotics administered, data needs to be presented on what percentage of antibiotics were considered "appropriately used'. Otherwise, the data given is just a list of antibiotics used in the different hospitals, without any clinical value to the readers. 
  2. There should be a discussion on why more antibiotics were common for CAIs than HAIs, because usually, HAIs are caused by bacteria with multidrug resistance. 
  3. The authors use the word "overuse". However, what is the definition of overuse in this study? Because in order to evaluate the "overuse" of antibiotics, appropriate indication and duration need to be considered. 

Minor comments: 

  1. line 43-44 "More than 700,000 people die from AMR, which will increase to 10 million by 2050..." --> Does this mean globally? Also, a reference needs to be added.

Reviewer 2 Report

Thanks very much for this opportunity to review this manuscript. This study applied the World Health Organization’s standardized point prevalence survey methodology to assess antibiotic use in 13 public and private not-for-profit hospitals in Uganda, covering 1,077 patients and 1,387 prescriptions. This study presented the status quo of antibiotic use in hospitals in Uganda, which could further be served as a baseline for global comparison and further quality improvement for the country. However, there are some main concerns that authors may need to address to help readers correctly interpret the results.

Major concerns:

Background:

  1. Line 44. The authors stated that “Increases in AMR due to the COVID-19 pandemic”. To my knowledge, it seemed that this statement has not been confirmed. Pls, check the reference to revise this statement.

Reference:

Monnet, Dominique L, and Stephan Harbarth. “Will coronavirus disease (COVID-19) have an impact on antimicrobial resistance?.” Euro surveillance : bulletin Europeen sur les maladies transmissibles = European communicable disease bulletin vol. 25,45 (2020): 2001886. doi:10.2807/1560-7917.ES.2020.25.45.2001886

  1. Line 55-56. “However, a key challenge … within health facilities”. What is the meaning of this sentence? Did you mean the consumption of antibiotics along with the distribution, including manufacturer, wholesalers, pharmacy outlets, hospitals, and primary cares? If so, the current study cannot address this issue and this statement seemed irrelevant to this study?

  1. It seemed that the authors missed the introduction of the status quo of antibiotic use and AMR in Uganda to help readers understand the settings of the current study. In addition, is there any study conducted in Uganda trying to understand the reasons for physicians’ irrational use of antibiotics? This information may need to be summarized to help readers understand the status quo of research relevant to the topic.

  1. The authors stated that the current study is belonging to ongoing continuous quality improvement plans, what are they? Is it necessary to briefly introduce the whole plan to help readers understand the setting and position of the current study?

Methods:

  1. To help readers understand the sampling process and how the facilities were chosen. There should be a summary of the Uganda healthcare system. What are the main providers of healthcare services in Uganda, what are the proportion of them, etc? In addition, information regarding how the sample size was calculated should be added.

  1. Though authors stated that they conducted the current study based on WHO PPS methodology. It’s still necessary to present the instruments applied in the current study. What are they and what variables were collected within each instrument? Is there any adjustment and how about the reliability and validity of these instruments? What are the independent variables and dependent variables and how they were calculated?

  1. In the data analysis part, authors still need to present the details of the data cleaning process of the current study, particularly 1) how antibiotics were defined; 2) how diagnose were classified, and 3) how the compliance was defined and calculated?

  1. It seemed that the authors only conducted the univariate analysis. Is it possible to further conduct a multivariable regression analysis, controlling the confounding variables, and give a more accurate estimation of the effect of factors on antibiotic use?

  1. It seemed that Figure 1 should be inserted in the methods part to be corresponding with the sampling process.

Results:

  1. It seemed the presentation of results is in a mess, with patient-level data inserted between the overall data. I recommended the authors re-arrange the results in the following order. First, present the overall antibiotic use data (2.3-2.5 & 2.7-2.10), follow by the hospital level data (2.6), and finalize by the patient-level data (2.1-2.2).

  1. Pls use graphic symbols to represent the AWARE category in Figure 2 rather than words to enhance visualization.

  1. Line148-149. It seemed that it’s not clear how the chi-squared test is calculated. A supplementary table presented the ceftriaxone prescription and the diagnosis would be clear. (Column & row)

  1. Line 154-156 & Line 159-160. Pls present the test applied and the results of the tests as well.

Discussion.

  1. Line 243. It seemed unclear that why the authors attributed the multiple doses of antibiotics to the lack of guidelines on antibiotic use in surgery. According to existing studies, multiple doses may due to incentives, insufficient knowledge, wrong attitudes, patient expectation, and also the lack of clinical guidelines. How did the authors consider the other factors that contributed to the results and why authors thought that the lack of clinical guidelines in SURGERY is a major concern?

  1. Line 250-252. It seemed that physicians wrote the reasons for antibiotic prescriptions in the current study. Is it possible to present these results to help us understand why physicians irrational prescribe or overuse antibiotics?

  1. Line 268-272. Similar as the comment 1 in the discussion. The prescribing behaviors could be complex, not only based on the policy and implementation of the clinical guideline. It has been shown that irrational antibiotic prescribe (non-compliance to the guideline) can be attributed to patient expectation, physician’s wrong attitudes (to comply with patients, fear of adverse events, etc.). Authors may need to refer to existing studies in Uganda, compare the results of the current study within the existing literature to find the reasons underlying the low compliance. In addition, without presentation of detailed information of how the compliance rate was calculated. We cannot conclude that whether the calculation of compliance may contribute to the low rate in Uganda.

Reviewer 3 Report

The Manuscript is well written and relevant to the Journal

  1. In the abstract include the key areas identified in the conclusion
  2. Introduction: What is the overall Antibiotic consumption in Uganda as per WHO report or other report: Is the consumption over or underestimated?  Namugambe J.S et al 2021, Trop. Med. Infect Dis
  3. Line 72: Report IQR when median is reported 
  4. Line 75-76: List the underlying conditions in descending order. All should be listed
  5. In Table 3: There is no need to list all hospitals(number are small for each), the grouping of public or private is enough
  6. In the discussion clearly discuss why males are  more given antibiotics than females: Issue of Gender and AMR, include report to justify  the observation or recommend further studies 
  7. In the methods: Clearly show how community or hospital infections were defined with approporiate references 

Reviewer 4 Report

General comments

There is inconsistency and interchangeably use of the terms antimicrobials and antibiotics. Though the title and objectives refers to antimicrobials, throughout the Result Section, the term antibiotic is used. These two terms might not mean the same. The authors should be consistent in the use of the correct terms!

Table 1

It is not clear as to why some of the hospitals are named “Kumi Hospital” others not!

Figure 1

The use of the bubble size indicate the number of hospital bed is confusing. It is difficult to make a difference between the size of the bubbles in the map!

Results (From Line 179)

It is not clear why some of the “n” are italicised, while others are not!!

Study limitation

This section is unnecessarily too long. The authors should consider summarizing the section.

Round 2

Reviewer 2 Report

Thanks very much for the revision. I have no further main concerns.

Minor issue: Pls re-order the results section, in which 2.3 was missed.